# An Expanded Landscape of Unusually Short RNAs in 11 Samples from Six Eukaryotic Organisms

**DOI:** 10.3390/ncrna8030034

**Published:** 2022-05-19

**Authors:** Marine Lambert, Sara Guellal, Jeffrey Ho, Abderrahim Benmoussa, Benoit Laffont, Richard Bélanger, Patrick Provost

**Affiliations:** 1CHU de Québec-Université Laval Research Center/CHUL Pavilion, 2705 Blvd. Laurier, Quebec City, QC G1V 4G2, Canada; marine.lambert@mail.mcgill.ca (M.L.); sara.guellal@crchudequebec.ulaval.ca (S.G.); jeffreyho1211@gmail.com (J.H.); abderrahim.benmoussa@umontreal.ca (A.B.); laffont.benoit@gmail.com (B.L.); 2Department of Microbiology, Infectious Diseases and Immunology, Faculty of Medicine, Université Laval, Quebec City, QC G1V 0A6, Canada; 3Department of Plant Sciences, Faculty of Agricultural and Food Sciences, Université Laval, Quebec City, QC G1V 0A6, Canada; richard.belanger@fsaa.ulaval.ca

**Keywords:** RNA sequencing, small RNA, unusually short RNA, non-coding RNA

## Abstract

Small RNA sequencing (sRNA-Seq) approaches unveiled sequences derived from longer non-coding RNAs, such as transfer RNA (tRNA) and ribosomal RNA (rRNA) fragments, known as tRFs and rRFs, respectively. However, rRNAs and RNAs shorter than 16 nt are often depleted from library preparations/sequencing analyses, although they may be functional. Here, we sought to obtain a complete repertoire of small RNAs by sequencing the total RNA from 11 samples of 6 different eukaryotic organisms, from yeasts to human, in an extended 8- to 30-nt window of RNA length. The 8- to 15-nt window essentially contained fragments of longer non-coding RNAs, such as microRNAs, PIWI-associated RNAs (piRNAs), small nucleolar RNAs (snoRNAs), tRNAs and rRNAs. Notably, unusually short RNAs < 16 nt were more abundant than those >16 nt in bilaterian organisms. A new RT-qPCR method confirmed that two unusually short rRFs of 12 and 13 nt were more overly abundant (~3-log difference) than two microRNAs. We propose to not deplete rRNA and to reduce the lower threshold of RNA length to include unusually short RNAs in sRNA-Seq analyses and datasets, as their abundance and diversity support their potential role and importance as biomarkers of disease and/or mediators of cellular function.

## 1. Introduction

Studies in the 1990s related to small non-coding RNAs (sRNAs) [1,2] were scarce and often defined them as junk RNAs, non-functional RNAs, or degradation products [3]. This dogma was firmly anchored in scientific debates, leading to a span of half a century between their first observation and the discovery of functional 19 to 24 nucleotide (nt) microRNAs (miRNAs) [4,5]. The latter discoveries have prompted researchers to look beyond and to challenge currently accepted beliefs and dogma.

Nowadays, several sRNA families, such as PIWI-associated RNAs (piRNAs) [6], small nucleolar RNAs (snoRNAs) [7], or cytoplasmic RNAs (Y RNAs) [8] have been reported and their functions deciphered. Among them, the control of gene expression [9], the induction of RNA chemical modifications [10], or the maintenance of genome integrity [11] have been demonstrated.

Recently, a plethora of studies, made possible by the use and the development of high-throughput sequencing (HTS) [12], have identified small RNA fragments derived from longer RNAs [13,14]. Accumulating evidence suggests that these RNA fragments are themselves physiologically relevant in health and disease [15,16]. Those were named according to the RNA from which they derived: transfer RNA (tRNA) derived fragments (tRFs), snoRNA-derived RNAs (sdRNAs) [17], and ribosomal RNA fragments (rRFs) [18]. Although nothing is known regarding Y-RNA fragments (YsRNA) function [19], tRFs have been described as regulators of gene expression [20,21], whereas sdRNAs are involved in the regulation of alternative splicing events [22], messenger RNA (mRNA) stability [23], or translation [24].

Small RNA sequencing (sRNA-Seq) approaches are designed to determine sRNA sequences and their relative abundance, which depend on the method used for library generation [25,26]. For the past decade, sRNA library preparation has involved a routine, standardized size selection of sRNAs by gel excision according to their length, and it essentially centered on that of microRNAs, i.e., ranging from 16 to 30 nt [27]. sRNAs shorter than 16 nt are thus systematically excluded from sRNA-Seq studies, the reason being that these sequences are too short to be functional or mapped reliably to the genome, and they unnecessarily increase the background noise, blurring downstream analyses of more important sequences.

Although some sequences shorter than 16 nt are degradation products, several might still be functional. Our laboratory previously reported the serendipitous discovery of a 12 nt semi-microRNA (smiRNAs) able to prevent the repression induced by the microRNA from which it originated [28], suggesting that unusually short RNAs may conceal interesting functionalities and that their study may shed light on new regulatory sequences. In addition, by-products of stable degradation may be useful as diagnostic or prognostic markers in pathologies, as with YsRNAs in cancer for instance [8,19].

The limitations we are facing in small RNA research are those we impose on ourselves and the technologies we are using rather than the technologies themselves. To challenge the current dogma of restraining our view and interest to RNA species longer than 16 nt, we lowered the 16-nt threshold to 8-nt, which is still arbitrary, while keeping the upper 30-nt limit.

The focus of the present study was to gather evidence about the differential enrichment of the small RNAs in the 8- to 30-nt range and to further highlight the existence of 8- to 15-nt RNA species, by sRNA-Seq analysis of 11 different biological samples from 6 different species, from yeast to human, thereby providing an unprecedented view of unusually short RNAs in eukaryotes.

We have pursued the analysis of some of the results we have obtained and recently reported the identification of a new family of unusually short RNAs mapping to ribosomal RNA 5.8S, which we named dodecaRNAs (doRNAs), according to the number of core nucleotides (12 nt) their members contain [29].

We hope that sharing our approach, findings, and complete datasets of eukaryotic RNA species between 8 and 30 nt in length, especially those in the 8- to 15-nt window, will stimulate research in that new area of unusually short RNAs.

## 2. Results and Discussion

### 2.1. Identification of Highly Abundant Unusually Short RNAs

Analysis of the reads distributed by sequence length unveiled new RNA abundance peaks below the standard 16-nt threshold. We observed a major peak at 12 and/or 13 nt, in all analyzed samples, except in *A. thaliana* (Figure 1A–E). In human samples, the average of 70,000 RPM was observed for the 12-nt RNAs and 350,000 RPM for the 13-nt RNAs, so the 13-nt RNAs were ~5 times more abundant than those of 12 nt. The same abundance difference was seen in yeast samples (Figure 1A,E). Equivalent abundance of 12- and 13-nt sequences was observed in mouse (Figure 1B) and plant (Figure 1D) samples. The opposite trend was seen in *D. melanogaster*, in which 12-nt RNAs (437,760 RPM) were more abundant than 13-nt RNAs (155,602 RPM; Figure 1C).

Moreover, we noted small peaks of 16-nt plant sequences (25,412 RPM; Figure 1D), of 18-nt mouse sequences (Figure 1B), and major peaks of 17- and 18-nt in human samples (Figure 1A). These latter peaks remained 2.4-fold smaller than 13-nt RNAs. In addition, human 12-nt RNAs were (from 20% to 80%, depending on the sample) less abundant than 17- or 18-nt RNAs, except in HUVEC, where they were 1.5 times more abundant than the 17-nt RNAs (Figure 1A). Although 13-nt RNAs in HUVEC and HEK293 cells were 6.7 and 5.1 times more abundant than 17-nt RNAs, those 17-nt RNAs were more abundant than 13-nt RNAs in human PMN (Figure 1A).

The 17- to 19-nt peak (Figure 1) observed in human samples is also present in mouse samples, albeit at a much lower level. These 17- to 19-nt long RNAs mainly comprise sdRNAs, tRFs, and rRFs (Figure 2 and Figure 3). In human samples, the 17-nt RNAs seem to be constituted of a unique sequence originating from *SNORD30* (Table 1). It was demonstrated in 2009 that snoRNA-derived RNAs (sdRNAs) from C/D snoRNAs exhibit a bimodal length distribution at ~17–19 nt and >27 nt and predominantly originate from the 5′ end [30]. A possible function of sdRNAs would be to act as a novel source of miRNAs, as a sdRNA originating from the snoRNA *ACA45* was found to co-immunoprecipitate with AGO1 and AGO2 in human embryonic kidney 293 cells (HEK293) [23]. SnoRNAs are also a reported source of piRNAs [31], but their mechanism of action and other of their functions remain unclear. Our data are consistent with 17- to 19-nt RNA species being produced by a processing mechanism preserved in humans and in mice, but not in other organisms, which warrants further investigation.

Assessment of RNA sequence diversity through the analysis of unique sequences revealed that the relatively abundant 12- and 13-nt sequences were not correspondingly diverse, as shown by their relatively lower content in unique sequences (Appendix A). Indeed, the number of unique 12- and 13-nt RNA sequences, yet overly abundant, in human, mouse, fly, and budding yeast (Appendix A), was lower than that of 18- and 22-nt RNAs, except in fission yeast.

The presence of two flattened peaks centered around 13- and 22-nt unique RNAs in fission yeast *S. pombe* (Appendix A), which are sharper in mouse (Appendix A) and human (Appendix A) samples, is worth noting from an evolutionary perspective.

### 2.2. RNAs Shorter Than 16- nt Are More Abundant in Bilaterian Organisms

We performed our analyses by comparing sRNAs obtained in the standard window of RNA length (i.e., 16- to 30-nt) to unusually short sRNAs found in the 8- to 15-nt window, both in terms of sequence and relative abundance. Even if the first window (i.e., 16- to 30-nt) was twice as large as the smaller (i.e., 8- to 15-nt) and contained the well-known family of microRNAs, we found a higher abundance of RNAs in the 8- to 15-nt window in mouse and Drosophila samples (Figure 1G,H). This was not the case for every human sample, except for HUVEC; in HEK293 cells, the reads abundance was almost equally distributed between the two windows, whereas in PMN the ratio of reads within the 8- to 15-nt window was slightly lower (Figure 1F). Finally, a higher proportion of reads within the standard window was found in *Arabidopsis thaliana* and *Saccharomyces cerevisiae* (Figure 1I,J), while an equal distribution between the two windows was observed in *Schizosaccharomyces pombe* (Figure 1J).

Although the number of samples we analyzed is small, it seems that in bilaterian organisms unusually short RNAs are particularly conserved and more abundant than sRNAs found in the window of 16- to 30-nt standard-length, while the opposite trend was observed in yeasts and plant. However, we did not observe this pattern in terms of diversity, as represented by the number of unique reads (Figure 1F–H vs. Appendix A).

We then performed analyses of conserved RNA sequences between the bilaterian species (human, mouse, and fly samples), as they might give some indications of their functionality. We obtained 181, 46, and 28 sequences shorter than 15-nt from conserved rRNAs, tRNAs, and microRNAs, respectively (Appendix A). The conserved cleavage and the generation of these sequences makes them particularly interesting, supporting the functionality of RNAs shorter than 15-nt.

### 2.3. Small Non Coding RNA Distribution upon Biotypes

Sequences obtained by sRNA-Seq were first annotated by mapping against downloaded databases of defined sRNA classes, including, among others, rRFs, tRFs, snoRNAs, microRNAs, and the piRNAs, providing an unprecedented, systematic analysis of RNA species from six eukaryotic organisms and between 8- and 30-nt in length.

#### 2.3.1. MicroRNAs Are Less Abundant Than Unusually Short RNAs

As observed in the nearly 2000 small RNA libraries listed by Li et al. [32], analysis of small RNA length distribution in the standard 16- to 30-nt window consistently showed the greatest abundance of RNAs in the 19- to 24-nt range, usually associated to microRNAs in human, mouse, or fly [33,34,35]. However, when reducing the lower threshold of the window to 8 nt, the 19- to 24-nt RNAs became a minority of all sequences and were largely dominated by shorter sequences (Figure 1A–D), suggesting that 19- to 24-nt sequences are not the most abundant sRNAs.

However, a high level of RNA diversity was observed in the 19- to 24-nt range, as reflected by the large number of unique sequences found among all species in which microRNAs are found (Appendix A). Thus, broadening of the standard 16- to 30-nt window to the 8- to 30-nt range appears to have reduced the relative abundance of 19- to 24-nt RNAs, while retaining their relative diversity.

Our data suggest that, although the preparation of an 8- to 30-nt library may not be the best to study only 19- to 24-nt RNAs, such as microRNAs, it allows a comparison of their relative abundance versus the other sRNAs detected, including their derived fragments, such as semi-microRNAs for microRNAs. Moreover, as these sequences are more abundant than microRNAs, it may facilitate their detection and use as biomarkers. However, their length poses a challenge to their rapid detection and their accurate quantitation.

#### 2.3.2. MicroRNA Fragments May Be Detected in the 8- to 15-nt Window

The microRNA and their fragment reads were almost not detectable in the 8- to 15-nt window (Figure 2A–C), while they constituted a significant part of 16- to 30-nt RNAs (Figure 2F–H).

Length distribution plotting of the reads mapping to microRNA sequences unveiled a large abundance of 19- to 24-nt RNAs (Figure 3A) that was associated with a high diversity (Appendix A), corresponding to the known and well-defined class of microRNAs. Nevertheless, a high diversity of 8- to 11-nt sequences (Appendix A, in blue) of relatively high abundance (Figure 3A) was also observed, which may be interesting, since they may be functional such as semi-microRNA [28]. Interestingly, our current analyses revealed the 10 most abundant of 8- to 15-nt RNA fragments derived microRNAs in all samples (Appendix A). The origin of each microRNA fragment seems to be conserved between different human samples and between mouse samples. Most of the microRNA fragment sequences are conserved between human and mouse samples, whereas a minority of sequences are conserved between human, mouse, and fly samples (Appendix A).

In total, we discovered between 886 and 3705 unique microRNA-derived sequences in the 8- to 15-nt window, depending on the sample (Appendix A). Listing the most abundant microRNA-derived fragments (Appendix A), we have discovered a fragment derived from miR-625 that can suppress the inflammatory response [36] in human PMN and HEK293 cells, but also in HUVEC with a lower RPM number. We also reported semi-microRNAs from miR-207, miR-1b, or miR-382 in mice, from miR-12136 involved in drug resistance [37], and from miR-221 or miR-1283 in humans (Appendix A). The major pathogenic role of microRNAs in cancer and inflammatory diseases has been widely described, as well as their use, on a clinical level, either for diagnosis or prognosis in pathologies [38]. In this context, miRNA fragments may have relevant gene regulatory functions, in addition to their potential utility as biomarkers, which is why the study and the monitoring of their expression need to be intensified to challenge the dogma about their uselessness. Indeed, even though the inhibitory principle of smiRNAs has been demonstrated only for a smiRNA-223, we speculate that smiRNA modulates the function of the microRNA from which they derive. Thus, these sequences may offer alternative therapeutic options for the treatment of pathologies [39].

#### 2.3.3. piRNA Fragments Are as Abundant and Diverse as piRNAs in Bilaterian Organisms

piRNAs are a class of small non-coding RNAs of 24- to 30-nt initially identified as repressors of germ cell transposition [40,41], and subsequently detected in somatic cells of many animals [42]. Recently, 20-nt long piRNA have been detected [43]. Interestingly, in this study, RNA shorter than 20-nt long piRNA were also detected and were ever more abundant than those of 20- and 24-nt in all samples. Since the functionality of piRNAs as biomarkers for early diagnosis in pathologies has been increasingly pointed out in recent years [41], as well as the function of small non-coding RNA from longer RNAs (e.g., tRFs, rRFs) [18,44], we aimed to identify piRNA-derived sequences in the 8- to 30-nt window.

Initially, the piRNAs and the piRFs class get the most abundant part in *Mus musculus* (Figure 2B,G), while the second species which has shown the most abundant piRNAs and piRFs is *D. melanogaster* (Figure 2C,H). Finally, in human samples the share of piRNAs is the lowest (Figure 2A,F). It seems that the share of piRNAs in the RNAs found in the 8 to 30-nt window seems to decrease from Drosophila to Human. In contrast to their abundance, we noticed that the fly piRNAs/piRFs were very diverse, since they have the largest share of unique sequences in our analyses (Appendix A). They account for more than 27.5% and 9.4% of unique sequences of unusually short RNAs (8–15 nt) and standard RNAs (16–30 nt), respectively, in human, mouse, and fly samples (Appendix A).

Moreover, the length distribution and the abundance clustering of the piRNA and piRF reads (Figure 3B) highlighted a blue cluster corresponding to the highest abundance RNAs. One part of this cluster has grouped RNAs from 22- to 24-nt, which is the usual length of piRNAs, while another part contains sequences of 12- to 19-nt in length (Figure 3B). This last RNAs range, which was likely missed in most other studies, suggests that piRFs may be as abundant as authentic piRNAs.

Analysis of piRNA/piRF diversity showed a large number of unique sequences in the 11- to 25-nt window, with a particular enrichment in 13-nt sequences across samples (Appendix A).

Then, we generated a list of the most abundant piRNAs and piRFs found in our samples (Table 2). We observed that piRF ranging from 23- to 28-nt in length were the most abundant (particularly those originating from piR-hsa-145507 in human samples and piR-mmu-25873647 in mice), as well as 13- to 22-nt piRFs. Interestingly, these piRFs always derived from the 5′ end of the original piRNA (whenever this information was available). For instance, we identified several piRFs of 15- or 18-nt, all originating from the 5′ end of piR-mmu-10912946, reminding the generation type of the tRF-5 [45].

Our findings support the existence of RNA fragments derived from piRNAs, similar to the many classes of RNAs derived from longer reference precursors. While current research is documenting the presence of piRNAs in somatic cells and trying to understand their role [41,43,46], we propose to take into account the possible existence of piRFs, which may bear some functionality and warrant further investigation. Furthermore, it would be highly informative to investigate whether these piRFs are derived from conserved piRNA clusters.

#### 2.3.4. Discovery of Highly Abundant, Diverse, and Unusually Short sdRNAs

sdRNAs are preferentially produced upon cleavage of the 5′ or 3′ end of snoRNAs [47], and their length varies between 20- and 30-nt [13,48]. Here, we quantitated reads mapping to annotated snoRNAs and assembled the sdRNA transcriptome of our samples in the 8- to 30-nt window (Figure 3C). As for piRNAs, sdRNA length distribution analysis identified three clusters of varying abundance, including sequences of 17- and 29-nt, the characteristic length of sdRNAs from C/D snoRNAs [30], which were particularly abundant (Figure 3C). In addition, we identified highly abundant 9- to 12-nt sdRNA sequences (Figure 3C) that also displayed the greatest diversity, with the largest number of unique sequences (Appendix A, cluster in blue).

Surprisingly, a single sdRNA derived from the 5′ end of snoR58C (or snoR58 in mouse) dominates the 29-nt peak, constituting between 45% and 94% of the reads (Table 1). Similarly, a single sdRNA sequence accounted for 99.9% of the reads within the 17-nt peak in all human samples, and another for 96% of the 17-nt reads in mouse samples (Table 1). These two sdRNAs derive from the 5′ end of human SNORD30 or mouse SNORD83B, respectively. Although a specific cleavage of the 5′ end of SNORD30 giving rise to a ~22-nt sdRNAs has been reported previously [13], the 17-nt SNORD30-derived sdRNA is more abundant than the 22-nt derivative. Thus, as SNORD30 expression has been correlated to tumor progression in several studies [49,50], the 17-nt derivative may constitute a better marker of immune cancers than the 22-nt form.

The three most abundant 9- to 12-nt sdRNAs detected in our samples are listed in Table 3. Intriguingly, these sdRNAs share a common ATGA motif at their 3′ end (Table 3).

Although these analyses do not confirm these sdRNAs to be functional or more than stable by-products, the preponderance of discrete unusually short sdRNAs in all of our samples, from yeasts to human, as well as their origin (e.g., the 5′ end of snoRNAs) in mouse, human, fly, and plant, or specific motif (e.g., ATGA), support some specificity in the biogenesis process and/or a conservation mechanism, both of which deserve further investigations.

#### 2.3.5. Detection of tRFs Shorter Than 16-nt

tRFs are a rapidly growing class of non-coding RNAs, as several studies have contributed to improve our knowledge of their expression [51], biogenesis [52], and function [53]. Notably, their discovery has paved the way to dismantle the dogma on the functionality of RNAs resulting from the degradation of longer non-coding RNAs, in addition to the rise of studies on rRFs [18]. Again, tRFs are being used to overcome dogmatic limits, here length, as a recent study in Arabidopsis reported a strong accumulation of 13- to 16-nt tRFs [51].

Our study expands on the latter by reporting the presence of tRFs in all of our eukaryotic samples across the 8- to 30-nt window of RNA length (Figure 2). tRFs are particularly abundant in the 15- to 18-nt range (Figure 3D), with a concomitantly high diversity of unique sequences in the same range (Appendix A).

To get further insight into their family affiliation, biogenesis or mode of degradation, we grouped tRFs according to their sequence and aligned them on the shortest, common seed sequence (Appendix A, tRFs). Within these files, we can observe what seems to be the normal process of tRNA degradation or very with low abundant sequences of decreasing size, in “staircase”. However, within these groups of tRNAs that appear to be digested, we observe very abundant tRFs. The latter could represent either by-products of degradation of stable tRNAs or tRFs. We also determine the most abundant tRFs detected in our sRNA-Seq analyses (Table 4). As for the longer tRFs already listed, we observed that tRFs shorter than 16 nt were mainly from the 5′ or 3′ end of tRNAs or from their loop [45,51]. Together, these clues suggest that tRFs shorter than 16 nt may not be mere degradation products, but functional tRFs that deserve to be studied further.

#### 2.3.6. rRFs Are Overly Abundant in Bilaterian Organisms

rRFs formed the most abundant class of RNA found in our sRNA-Seq datasets, representing between 16% and 95% of the reads in the 8- to 15-nt window, and between 9% and 70% of the reads in the 16- to 30-nt window (Figure 2). Expansion of the sequencing window to include 8- to 15-nt RNA species considerably increased the share of rRFs, as compared to the standard 16- to 30-nt window, at the expense of all other biotypes in bilateral organisms (Figure 2A–C,F–H). The excessive abundance of rRFs, which has been observed by other groups [54,55] and discussed previously by us [3], suggests that they originate from rRNA, which constitutes 80% of the RNAs in cells [56].

As we did with tRFs, we gathered all rRFs having a common motif or sequence together (Appendix A, rRFs). Thus, a staircase-like profile with a correspondingly low read count for each rRF sequence emerged, supporting their active degradation, most often from 5′ to 3′, possibly involving exonuclease(s). sRNA-Seq analysis in an extended 8- to 30-nt window may thus provide, besides a more global, unbiased view of small RNAs, a means to better characterize a small RNA degradome and to study the mechanisms involved [12,57].

Some rRF sequences exhibited a peculiar profile of abnormally high read counts (Appendix A, rRFs, highlighted in blue), which did not correspond to the shortest reads, arguing against rRFs accumulating as degradation end-products of a stepwise nucleotide excision process. Listing of the most abundant rRFs revealed that the human and mouse sequences (except for one) shared the same minimal 12-nt sequence, with identical 5′ extension of mainly 1 (a C) or 6 nucleotides (TCGTAC) (except for one).

Heatmap analysis of rRF abundance according to their length confirmed the high abundance of 12-, 13- and 18-nt sequences (Figure 3E, clusters in blue). Each of these peaks was almost exclusively composed of a single sequence identical to those listed in Table 5: the sequence GACTCTTAGCGG represented 98% of the 12-nt reads, CGACTCTTAGCGG 95% of the 13-nt reads, and TCGTACGACTCTTAGCGG ~90% of the 18-nt reads in mouse and human samples (5′ extensions are underlined).

### 2.4. RT-qPCR Validation of Two Unusually Short rRFs of 12 and 13 nt

Analysis of our sRNA-Seq data revealed that in HEK293 cells the specific 12- and 13-nt rRFs represented 57,921 RPM and 338,156 RPM, respectively (~5 times more 13-nt sequences than the 12-nt), while miR-25 and miR-30a represented only 257 and 96 RPM, a ~3-log difference (Figure 4A). In mouse neuronal N2a cells, the two rRFs exhibited similar levels of expression, with 391,832 RPM for the 12-nt rRF and 349,502 RPM for the 13-nt rRF, while miR-25 and miR-30a accounted for 1220 and 731 RPM, respectively. The other human and mouse samples showed a similar relative distribution (Figure 4A). The predominance of the specific 12- and 13-nt rRFs among our sRNA reads prompted us to validate their existence by an alternate method (e.g., RT-qPCR), which would also allow a comparison of their abundance relative to better-known microRNAs.

We have thus developed a specific and sensitive RT-qPCR method based on splinted 5′ ligation to detect two particularly abundant rRFs of 12 and 13 nt that differ by only a single nucleotide at the 5′ extremity [58]. We used our new method to validate the existence of the two 12- and 13-nt rRFs, and quantitate them relative to two microRNAs: miR-25, which is highly expressed, and miR-30a, which is less expressed in the same human and mouse samples.

RT-qPCR analyses of the same human and mouse samples that were previously analyzed by sRNA-Seq confirmed the existence of the 12- and the 13-nt rRFs, and their relative abundance to each other; the 13-nt rRF being 5 times more abundant than the 12-nt rRF in human samples (Figure 4B). More importantly, our RT-qPCR data confirmed our sRNA-Seq showing the far greater abundance of the 12- and the 13-nt rRFs relative to microRNAs miR-25 and miR-30a (Figure 4B). A comparison between unusually short, 12- to 13-nt rRFs and 19- to 24-nt microRNAs, either in number of reads, height of the peaks or copy numbers, suggests that rRFs are overly more abundant than microRNAs. These findings were confirmed by RT-qPCR analysis of samples from different cultures of human HEK293 and mouse N2a cells in which, again, the 13-nt rRF was 5 times more abundant than the 12-nt rRF in human samples, and 12- and 13-nt rRF copy numbers proved to be ~3 logs higher than the individual microRNAs miR-25 and miR-30a (Figure 4C).

Together, the results obtained through the use of our new RT-qPCR detection method, tailored to monitor unusually short RNA species, allowed us to confirm our sequencing results, document the differential enrichment of 12- and 13-nt rRFs between human and mouse samples, and highlight their ~3-log higher levels of expression compared to individual microRNAs.

### 2.5. Conclusions

Although the cellular role and function of the newly discovered, unusually short RNAs in this study have yet to be elucidated, our results support their presence as genuine, as confirmed and quantitated by RT-qPCR and as suggested by the accumulation of rRFs of discrete length in all the samples analyzed. Regardless as to whether these sRNAs are stable degradation by-products or fulfill a specific function within cells, their systematic removal from sRNA-Seq analyses definitely hampers their study and their characterization, and it contributes to perpetuating the concept that unusually short RNAs are devoid of interest and utility.

However, it is necessary to relativize our results with respect to the potential bias introduced during the construction of the sequencing library. Indeed, although some unusually short sequences seem to represent a majority share in our results (e.g., sdRNA, tRF, or rRF sequences represent 90% of the abundance of the peak length; Table 1 and Table 3), the presence of post-transcriptional modifications of RNA may hinder the detection of some RNA species by HTS, and thus bias the sequencing results. For example, tRNAs/tRFs are highly modified post-transcriptionally. These modifications impair their reverse transcription and thus the ability of tRNA-derived sequences to be sequenced [59]. Thus, these sequencing biases may result from the inability to detect specific sequences. In particular, this could explain our results in Figure 2, in which a low share of tRFs is observed in most organisms in the 16- to 30-nt window, while it is widely known that these are highly abundant small RNA species in these organisms. Similarly, it has been widely reported that human tRFs derived from tRNAGly are the most abundant tRFs in most HTS analyses (Table 4), but Northern blotting performed in a previous study revealed that they are not particularly more abundant than other tRFs [60]. This is, in part, because tRNAGly is less modified than other tRNAs.

As our studies are based on RNA sequencing without prior demethylation, it is likely that we mainly documented small RNAs from longer RNAs that do not show post-transcriptional modifications. Thus, if the shorter RNAs are not post-transcriptionally modified, they will be more likely to be sequenced, leading to the misinterpretation that they are more abundant than the 16- to 30-nt sequences. It would be interesting to perform a comparative analysis with sequencing from a constructed library taking into account this possibly major bias. Nevertheless, this bias does not call into question the discovery of unusually short sequences shorter than 15-nt. On the contrary, we may even discover a greater diversity of them by sequencing RNAs in the 8- to 30-nt window from previously demethylated RNAs.

Finally, since the discovery that several sRNAs derive from longer RNAs, their usefulness in the diagnosis and prognosis of various pathologies has been established [8,13,44,54], albeit without precisely knowing their function, making their potential use in targeted medical research promising. We, therefore, call upon the scientific community to reconsider the dogma at play—as we all did when our research led to the discovery of microRNAs—and to reduce the lower limit of RNA length to include these unusually short RNAs in sRNA-Seq analyses and datasets. Their abundance and their diversity support their potential role and importance as biomarkers of disease and/or mediators of cellular function [29]. As a matter of fact, the study by Li et al. [61], who described the presence of unusually short 17-nt RNAs that can modulate human gene expression, has laid the foundation for the sRNA functionality length threshold to be enlarged. Today, we again set an arbitrary limit at 8 nt, even knowing that future studies may end up revealing that RNAs shorter than 8 nt also have biological relevance.

## 3. Materials and Methods

### 3.1. Ethical Statement

#### 3.1.1. Human Blood Samples

Collection of venous blood from healthy volunteers (adult Caucasians of both sexes from the immediate region of Quebec City) was approved by our institutional human ethics committee (B14-08-2103). The participants provided their written informed consent to participate in this study, in accordance with the Declaration of Helsinki.

#### 3.1.2. Mouse Tissue Samples

This study was carried out in accordance with the guidelines, regulations, and requirements of the Canadian Council of Animal Care for Animals Used for Scientific Purposes. Experiments were performed in accordance with the latest guidelines and using a protocol approved by the Université Laval Animal Welfare Committee.

### 3.2. Biological Samples

#### 3.2.1. Primary and Cultured Human Cells

Human blood polymorphonuclear leukocytes (PMN) were isolated from venous blood collected from four healthy donors and pooled, as described in Laffont et al. [62]. Human umbilical vein endothelial cells (HUVEC; Stem Cell Technologies, Vancouver, BC, Canada) were cultured in endothelial growth medium (Lonza, Basel, Switzerland) supplemented with bovine brain extract (Lonza, Basel, Switzerland) and maintained at 37 °C under 5% CO_2_ and used between passages 2 to 6. Cultured human embryonic kidney 293 (HEK293; ATCC, Manassas, VA, USA) were maintained in Dulbecco’s modified Eagle’s medium (DMEM) supplemented with 10% (*v*/*v*) fetal bovine serum (FBS), 1 mM sodium pyruvate, 100 units/mL penicillin, 100 μg/mL streptomycin, and 2 mM L-glutamine in a humidified incubator under 5% CO_2_ at 37 °C.

#### 3.2.2. Primary and Cultured Mouse Cells and Tissues

Mouse blood PMN were isolated from four healthy 12- to 15-week-old mice, as described in Duchez et al. [63], and pooled. The brain cortex (cerebellum) was collected, after PBS washing, from exsanguinated 24-month-old C57BL/6 mice and flash frozen in liquid nitrogen before storage at −80 °C. Neuronal N2a and NIH/3T3 fibroblast cell lines used in this study were obtained from ATCC (Manassas, VA, USA) and maintained in culture according to ATCC’s recommendation.

#### 3.2.3. *Drosophila melanogaster*

Adult flies were purchased from the University of California in San Diego (UCSD) Drosophila Species Stock Center (San Diego, CA, USA) and snap-frozen in liquid nitrogen before storage at −80 °C.

#### 3.2.4. *Arabidopsis thaliana*

*Arabidopsis thaliana*, accession Colombia (Col-0) was obtained from the ABRC (Arabidopsis Biological Resource Centre, Ohio State University, Columbus, OH, USA). Seeds were sown in autoclaved Pro-Mix BX potting mix (Premier Horticulture, Rivière-du-Loup, QC, Canada), watered with sterile distilled water (300 mL/pot/week), and placed in a growth chamber (16 h light at 22 °C, 8 h dark at 20 °C, 60–70% humidity). After 7 days, seedlings of uniform size were transferred to pots containing Pro-Mix BX potting mix and grown for 14 days prior to being harvested for RNA extraction.

### 3.3. Total RNA Isolation

Total RNA samples were prepared using TRIzol^®^ reagent or TRIzol LS^®^ reagent for liquid samples (Invitrogen Life Technologies, Carlsbad, CA, USA), following the manufacturer’s recommendations. Contaminating DNA was degraded using DNase I (M0303S, New England Biolabs, Ipswich, MA, USA) treatment.

### 3.4. Small RNA Library and Sequencing

For primary blood PMN samples involving cell isolation, we used a pooling strategy involving equal mixing of total RNA samples derived from independent biological samples. Pooling of small RNA samples is effective in reducing data variability, and it reduces the number of replicates, hence lowering the cost for subsequent steps [64]. For each of the other samples, a unique total RNA specimen was analyzed.

The quality and the concentration of the total RNA samples were verified using a NanoDrop ND-1000 (Thermo Fisher Scientific, Waltham, MA, USA) and gel separation (Appendix A). The sRNA-Seq libraries, containing RNA species between 8 and 30 nt in length were prepared as previously described [65]. Samples were diluted to a final concentration of 8 pM, denatured as single-stranded DNA, and cluster generation was performed on the Illumina cBot using a TruSeq Rapid SR cluster kit (GD-402-4001, Illumina, San Diego, CA, USA). Afterward, the clusters were sequenced for 51 cycles on Illumina HiSeq 2000 using TruSeq Rapid SBS Kits (FC-402-4002, Illumina, San Diego, CA, USA), as per the manufacturer’s instructions.

### 3.5. Analysis Workflow

Clean reads matching the quality standards were processed to remove the adaptor sequence, leading to sRNA trimmed reads. All reports displayed here were generated through the standard analysis pipeline of Arraystar Inc., (Rockville, MD, USA) (https://www.arraystar.com/, accessed on 17 May 2022) and refined using R (Free Software Foundation). Only the reads that were identical, both in length and in sequence, were considered as a unique read. TPM normalizations were not based on cell types—the increased number of chromosome/gene copies may contribute to increase the abundance of sRNAs in the cancerous HEK293 and N2a cell lines compared to primary cells.

After adapter trimming, low-quality filtering, and contamination checking, the clean sequencing data were mapped to the downloaded sRNA databases of the corresponding organisms, as listed in Appendix A. The mapped tags were then used to identify and to profile sncRNAs, including miRNAs (and semi-microRNAs), rRFs, tRFs, sdRNAs, piRNAs (and piRFs), and other sncRNAs based on the constructed libraries (Appendix A).

Post-transcriptional modifications of RNA can lead to errors in reverse transcription, whereas the presence of single nucleotide polymorphism (SNP) may hamper proper identification. Since the sequenced sRNAs may contain mismatches, and to allow detection of modified RNA species, such as those derived from tRNAs, rRNAs or piRNAs, we proceeded with our analyses by accepting one mismatch. Many piRNA sequences are actually derived from tRNA, or rRNA genes [66]. To avoid misclassification of small RNAs derived from piRNA/tRFs/rRFs, we prioritized the search and the classification of sequences originating from tRNAs or rRNAs in the class of tRFs or rRNAs, before that of piRNA sequences. For tRF/rRF identification, the reads aligned to precursor tRNA/rRNA genes and mature tRNA/rRNA sequences with the same strand as the source tRNA/rRNA were used for tRF/rRF analysis. In addition, sequences that did not perfectly match to any database were listed as undefined. No mismatch or gap was tolerated in our classification. miRNA precursor sequences and other known small non-coding RNA sequences were retrieved from the latest miRBase database [67] (release v22.0).

### 3.6. Adapter-Ligated RT-qPCR Method

This specific quantification method was set up, experimentally validated, and described in Lambert et al. [58]. The splint and the adaptor were annealed together prior to being added to 300 ng of total RNA in a ligation reaction intended to extend two unusually short rRF sequences (named doRNA and C-doRNA) at their 5′ extremity. Two µL of total ligated RNA were used for reverse transcription using the miRCURY Locked Nucleic-Acid (LNA)-modified microRNA PCR Assay (Qiagen Inc., Toronto, ON, Canada). After cDNA dilution (1/10), qPCR was performed using miRCURY LNA SYBR^®^ Green PCR Kits (Qiagen Inc., Toronto, ON, Canada) in 96-well plates using the CFX96 Touch™ Real-Time PCR Detection System (Bio-Rad, Mississauga, ON, Canada) and specific Custom LNA Oligonucleotides (Qiagen Inc., Toronto, ON, Canada). doRNA, C-doRNA, microRNA-25 (miR-25), and microRNA-30a (miR-30a) copy numbers were determined with a standard curve built using synthetic RNAs (Integrated DNA Technologies, Inc., Coralville, IA, USA). In compliance with the MIQE guidelines, the small nucleolar RNA U6 was used as a reference gene for normalization [68].

## Figures and Tables

**Figure 1 ncrna-08-00034-f001:**
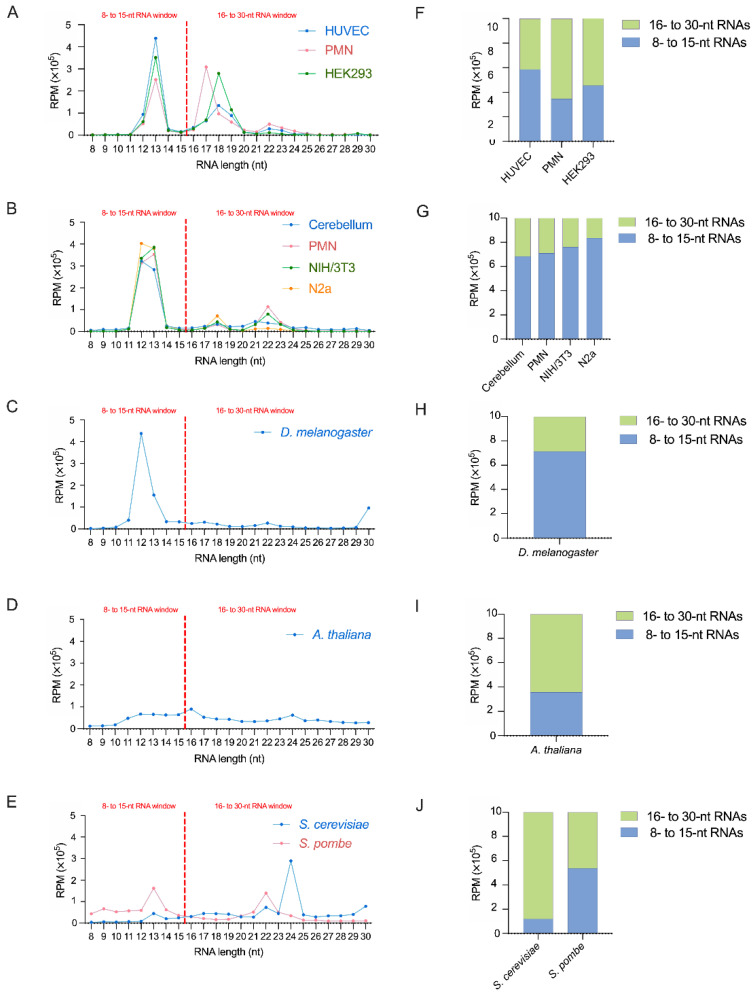
Opening of the 8- to 15-nt window revealed a high abundance of unusually short RNAs in human, mouse and fly samples. (**A**–**E**) Length distribution of the small RNA reads (in nucleotides, nt) from human (**A**), mouse (**B**), *D. melanogaster* (**C**), *A. thaliana* (**D**), *S. cerevisiae* and *S. pombe* (**E**) samples in the 8- to 15-nt and the standard 16- to 30-nt windows of RNA length. (**F**–**J**) Relative proportion of the reads in either of the two windows of RNA length in each sample.

**Figure 2 ncrna-08-00034-f002:**
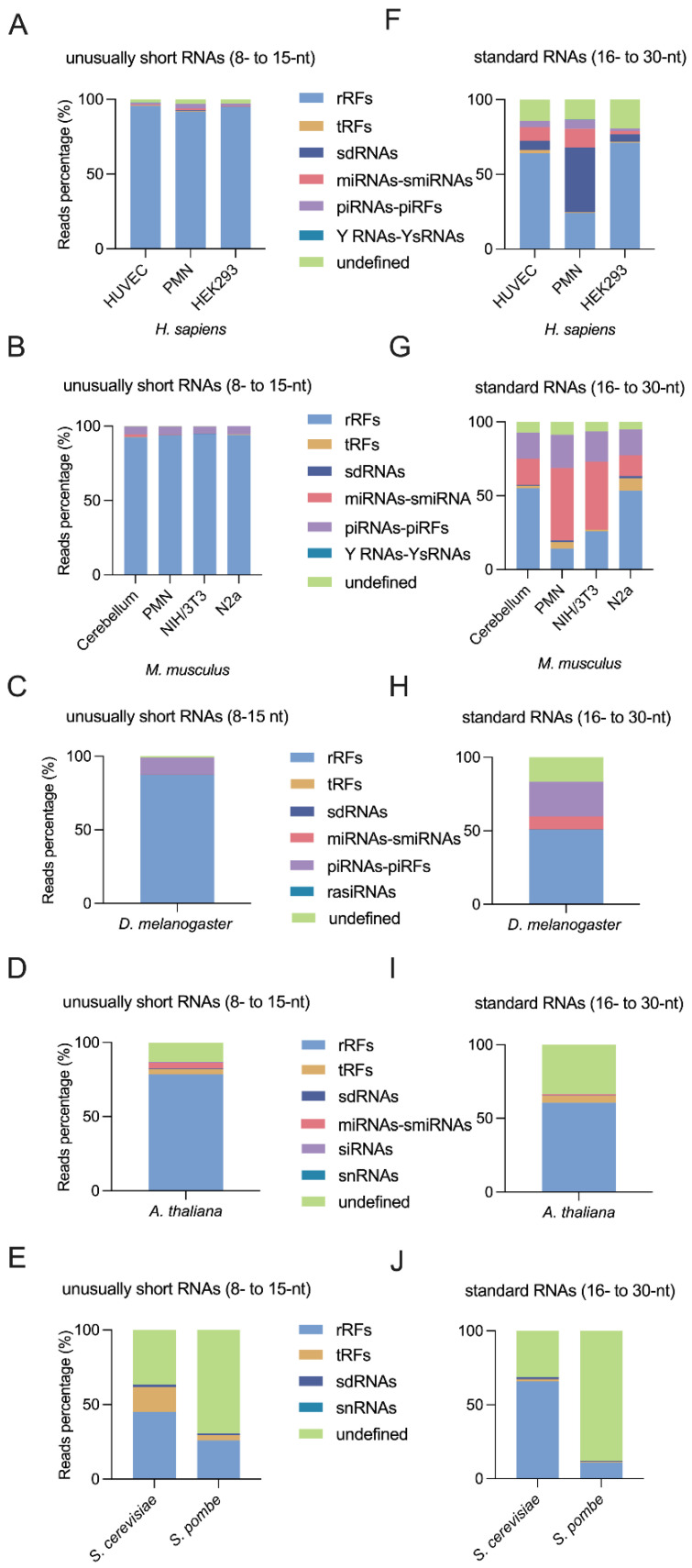
Most unusually short RNAs detected in the 8- to 15-nt window derive from rRNA in human, mouse, and fly samples. (**A**–**E**) Biotype distribution of the small RNAs (percentage of RPM) from human (**A**), mouse (**B**), *D. melanogaster* (**C**), *A. thaliana* (**D**), *S. cerevisiae* and *S. pombe* (**E**) samples in the 8- to 15-nt window. (**F**–**J**) The small RNA biotype distribution (percentage of RPM) of the corresponding samples in the standard 16- to 30-nt window.

**Figure 3 ncrna-08-00034-f003:**
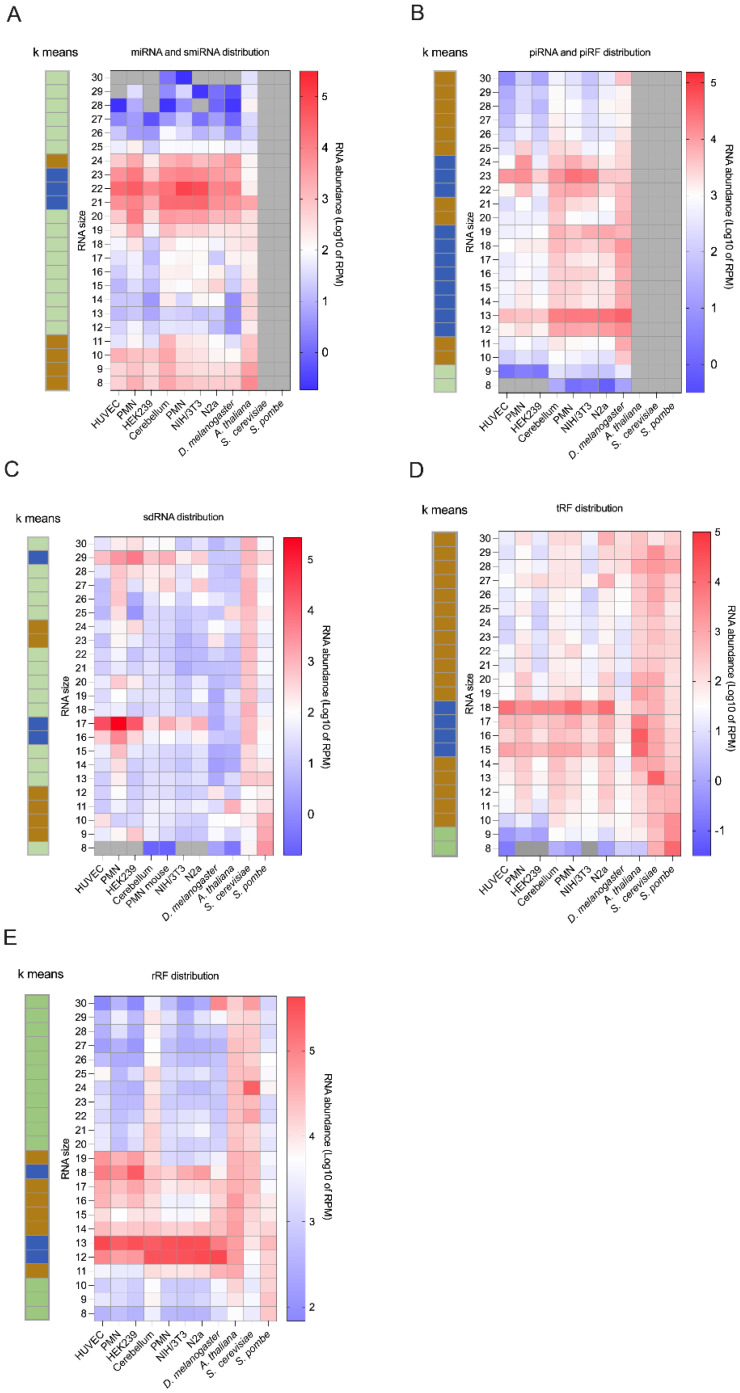
High abundance of fragments derived from longer, authentic non-coding RNAs detected by sRNA-Seq analysis in the 8- to 30-nt window. (**A**–**E**) Heatmap of length distribution abundance (log10 of RPM) of miRNA-miRNA (**A**), piRNA-piRF (**B**), sdRNA (**C**), tRF (**D**), and rRF (**E**) biotypes from human, mouse, *D. melanogaster*, *A. thaliana*, *S. cerevisiae* and *S. pombe* samples in the 8- to 30-nt window. Gray boxes display conditions with zero reads. For each biotype, a k-mean clustering was generated according to the small RNA length abundance distribution (Euclidean distance, *n* = 3 clusters; in blue the most abundant, while the green is the lesser abundant group and in beige were grouped RNAs with a middle abundance).

**Figure 4 ncrna-08-00034-f004:**
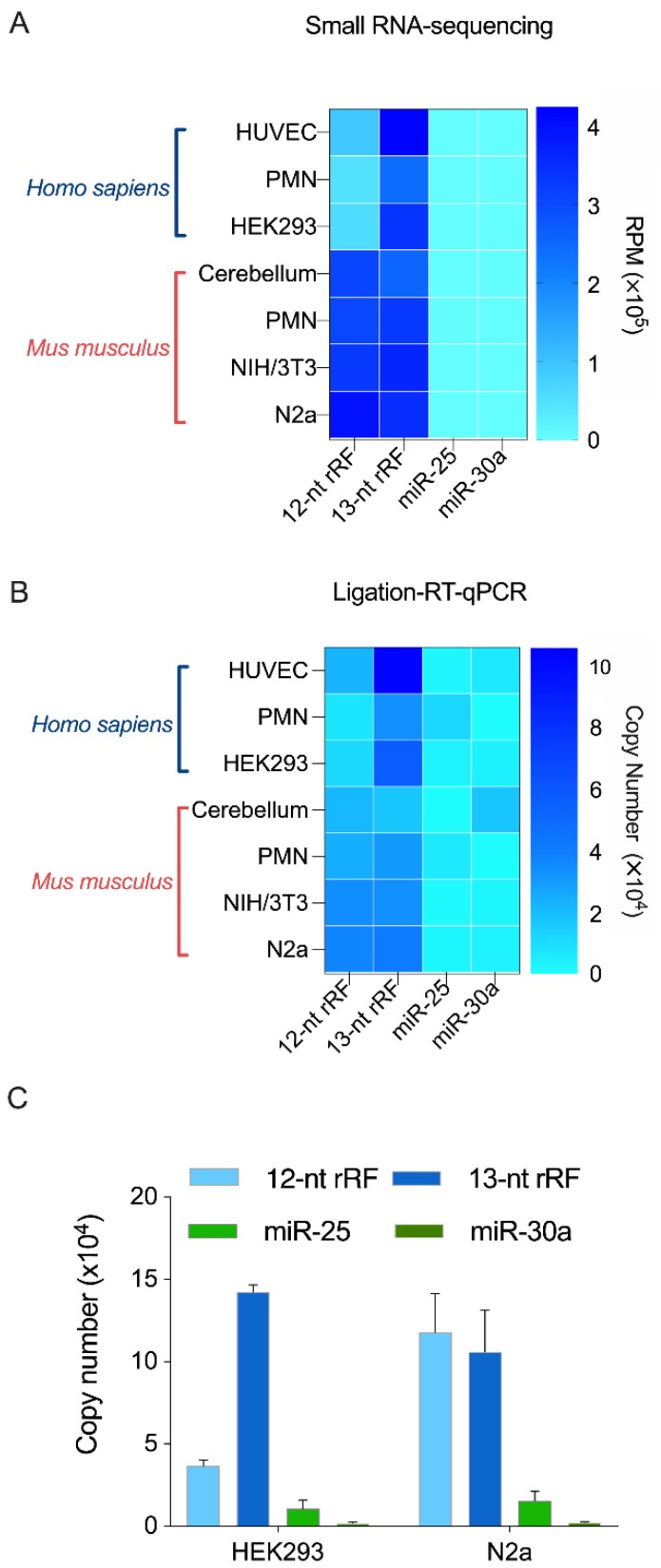
A 12-nt and a 13-nt rRF sequences are more abundant than microRNAs miR-25 and miR-30a in human and mouse samples. (**A**,**B**) Heatmap comparing the levels of 12-nt and 13-nt rRFs versus microRNAs miR-25 and miR-30a expressed either as reads per million (RPM) detected by sRNA-Seq (**A**) or in copy number detected by RT-qPCR (**B**). (**C**) Quantitation of 12 and 13-nt rRF levels by splinted ligation RT-qPCR analysis of total RNA extracted from HEK293 and N2a cells, in parallel to that of microRNAs miR-25 and miR-30a (*n* = 3 independent experiments). The detailed results of the statistical analyses are shown in Appendix A (two-way ANOVA with Holm Sidak’s post-hoc test).

**Table 1 ncrna-08-00034-t001:** Most abundant sdRNA sequences identified in the standard, 16- to 30-nt window by sRNA-Seq analysis.

		Length (nt)	Sequence	Reads *	Origin
*H.* *sapiens*	HUVEC	17	GTTTGTGATGACTTACA	99.9	5′ end of SNORD30
		29	TTGCTGTGATGACTATCTTAGGACACCTT	94.4	5′ end of SNORD58C
		29	CTGCAGTGATGACTTTCTTAGGACACCTT	4.4	5′ end of SNORD58A
	PMN	17	GTTTGTGATGACTTACA	99.9	5′ end of SNORD30
		29	TTGCTGTGATGACTATCTTAGGACACCTT	54.2	5′ end of SNORD58C
		29	CTGCAGTGATGACTTTCTTAGGACACCTT	29.7	5′ end of SNORD58A
	HEK293	17	GTTTGTGATGACTTACA	99.9	5′ end of SNORD30
		29	TTGCTGTGATGACTATCTTAGGACACCTT	94.6	5′ end of SNORD58C
		29	CTGCAGTGATGACTTTCTTAGGACACCTT	4.7	5′ end of SNORD58A
*M.* *musculus*	Cerebellum	17	GTTCTGTGATGAGGCTC	96	5′ end of SNORD83B, without the 3 first nt
		29	TTGCTGTGATGACTATCTTAGGACACCTT	64	5′ end of SNORD58, without the 3 first nt
		29	CTGCAGTGATGACTATCTTAGGACACCTT	17	5′ end of SNORD58, without the 3 first nt
	PMN	17	GTTCTGTGATGAGGCTC	98	5′ end of SNORD83B, without the 3 first nt
		29	TTGCTGTGATGACTATCTTAGGACACCTT	73	5′ end of SNORD58, without the 3 first nt
		29	CTGCAGTGATGACTATCTTAGGACACCTT	13	5′ end of SNORD58, without the 3 first nt
	NIH	17	GTTCTGTGATGAGGCTC	99	5′ end of SNORD83B, without the 3 first nt
		29	TTGCTGTGATGACTATCTTAGGACACCTT	47	5′ end of SNORD58, without the 3 first nt
		29	CTGCAGTGATGACTATCTTAGGACACCTT	35	5′ end of SNORD58, without the 3 first nt
	N2a	17	GTTCTGTGATGAGGCTC	99	5′ end of SNORD83B, without the 3 first nt
		29	TTGCTGTGATGACTATCTTAGGACACCTT	45	5′ end of SNORD58, without the 3 first nt
		29	CTGCAGTGATGACTATCTTAGGACACCTT	26	5′ end of SNORD58, without the 3 first nt

* % of reads from sequences having the same length. Nucleotide substitutions are in red.

**Table 2 ncrna-08-00034-t002:** Most abundant piRNAs and piRFs identified by sRNA-Seq analysis.

		Length (nt)	Sequence	Origin (piRBase Name)
*H. sapiens*	HUVEC	15	GACCAATGATGTGAA	piR-hsa-4433698 5′ end
		23	TCCTGTACTGAGCTGCCCCGAGA	piR-hsa-145507
		23	TCCTGTACTGAGCTGCCCCGAGT	piR-hsa-145507
	PMN	15	TACAACTTTTGGCAA	piR-hsa-7695930 3′ end
		14	ACAACTTTTGGCAA	piR-hsa-7695930 3′ end
		23	TATTGCACTTGTCCCGGCCTGTA	piR-hsa-137098
	HEK293	14	GATGGGTGACCGCC	piR-hsa-741077 fragment
		13	ATGGGTGACCGCC	piR-hsa-741077 fragment
		23	TCCTGTACTGAGCTGCCCCGAGA	piR-hsa-145507
*M. musculus*	Cerebellum	15	GCATTGGTGGTTCAG	piR-mmu-10912946 5′ end
		18	GCATTGGTGGTTCAGTGG	piR-mmu-10912946 5′ end
		23	AACCCGTAGATCCGAACTTGTGA	piR-mmu-29307247 5′ end
		23	TCCTGTACTGAGCTGCCCCGAGA	piR-mmu-25873647 5′ end
	PMN	15	GCATTGGTGGTTCAG	piR-mmu-10912946 5′ end
		16	AGCGGAGTAGAGCAGT	piR-mmu-23655655 5′ end
		23	TCCTGTACTGAGCTGCCCCGAGA	piR-mmu-25873647 5′ end
		23	TCCTGTACTGAGCTGCCCCGAGT	piR-mmu-25873647 5′ end
		22	CCTGTACTGAGCTGCCCCGAGA	piR-mmu-25873647 5′ end
		23	GTACCCTGTAGATCCGAATTTGT	piR-mmu-11542414
	NIH/3T3	16	AGCGGAGTAGAGCAGT	piR-mmu-23655655 5′ end
		22	CCTGTACTGAGCTGCCCCGAGA	piR-mmu-25873647 5′ end
		23	TCCTGTACTGAGCTGCCCCGAGA	piR-mmu-25873647 5′ end
		24	GTCCTGTACTGAGCTGCCCCGAGA	piR-mmu-25873647 5′ end
	N2a	12	TCGCTGTGATGA	piR-mmu-24106721
		23	CACCCGTAGAACCGACCTTGCGT	piR-mmu-31228201 5′ end
		27	GGCTCTGTGGCGCAATGGATAGCGCAT	piR-mmu-5102689
		28	TGGCCAAGGATGAGAACTCTAACCTGAC	piR-mmu-7884931
*D.* *melanogaster*		13	GAGGAAACTCTGG	piR-dme-108681 5′ end
		15	AAGGGAAGGGTATTG	piR-dme-5048778 5′ end
		16	AAAGGGAAGGGTATTG	piR-dme-5048778 5′ end
		18	CTGGGTCGGCCGGGGCGC	piR-dme-34359551 fragment
		20	TAGGGACGGTCGGGGGCATC	piR-dme-40694119 3′ end
		21	ATAGGGACGGTCGGGGGCATC	piR-dme-40694119 3′ end

**Table 3 ncrna-08-00034-t003:** Most abundant sdRNA sequences identified in the 8- to 10-nt cluster by sRNA-Seq analysis.

		Length (nt)	Sequence	Reads *	Origin
*H. sapiens*	HUVEC	9	GGCTAATGA	97.9	5′ end of SNORD-like-snoRNA,alias:ZL45, ID:snoID_0724, without the 3 first nt
		12	TCGCTATGATGA	36.9	5′ end of SNORD14B
		10	GGACCAATGA	96.0	5′ end of SNORD114-12
	PMN	9	GGCTAATGA	99.7	5′ end of SNORD-like-snoRNA,alias:ZL45, ID:snoID_0724, without the 3 first nt
		12	TCGCTATGATGA	23.9	5′ end of SNORD14B
		11	CCCGTCTGACC	22.0	3′ end of SNORD13
	HEK293	9	GGCTAATGA	100,0	5′ end of SNORD-like-snoRNA,alias:ZL45, ID:snoID_0724, without the 3 first nt
		10	TGGCTAATGA	45,2	5′ end of SNORD-like-snoRNA,alias:ZL45, ID:snoID_0724, without the 2 first nt
		11	GTAAGTATATT	41,4	Middle of SNORA24L2
*M. musculus*	Cerebellum	11	CGCTGTGATGA	32.6	5′ end of SNORD14C, without the first nt
		9	ATTGAGGAC	7.9	CD_40-1_ (chr16) 20684238,20684314
		12	AATTGTGGTAAC	13.6	Middle of SCARNA10
	PMN	11	CGCTGTGATGA	37.9	5′ end of SNORD14C, without the first nt
		12	AATTGTGGTAAC	8.3	Middle of SCARNA10
		11	ATTGTGGTAAC	11.3	Middle of SCARNA10
	NIH/3T3	11	CGCTGTGATGA	19.5	5′ end of SNORD14C, without the first nt
		11	AGAGAGGTGAG	18.1	Middle of SNORA17
		12	TGCTGTGATGAC	39.6	5′ end of SNORD58C, without the first nt
	N2a	11	CGCTGTGATGA	80.4	5′ end of SNORD14C, without the first nt
		12	AGGGATTGTGGG	28.2	5′ end of SNORA71
		10	GCGGGTGTGG	24.1	SNORA74B
*D.* *melanogaster*		12	GTGGAGGTAAAG	98.0	5′ end snoRNA:Psi18S-525f
		9	ATAGGGACG	71.3	snoRNA:Psi18S-525k (Dmel_CR34569)
		10	TTATAAACTG	43.7	PsiU2-38.40.42 (scaRNA:PsiU2-38.40.42)
*A. thaliana*		11	AGATATGATGA	95.1	5′ end of SnoR18a
		10	AATATTGAAA	31.4	Middle of SnoR96
		11	TAATATTGAAA	1.4	Middle of SnoR96
*S. cerevisae*		10	CCTTCTGAAA	22.1	SnoRNA86
		11	TCCTTCTGAAA	26.6	SnoRNA86
		10	TCGGGGCTGA	11.4	SnoRNA86
*S. pombe*		9	TCAACTGTA	28.0	SnR70
		10	TGTTCTGATG	35.5	SnR81
		9	TGTCTGATC	6.7	Snr41

* % of reads from sequence with the same length. Nucleotide substitutions are in red. The common motif at the 3′ end is highlighted in green.

**Table 4 ncrna-08-00034-t004:** Most abundant tRF sequences identified by sRNA-Seq analysis.

		Length (nt)	Sequence	Origin
*H. sapiens*	HUVEC	18	GCATTGGTGGTTCAGTGG	5′ end of tRNA-Gly-GCC-3-1
		18	GCATGGGTGGTTCAGTGG	5′ end of tRNA-Gly-GCC-3-1
		15	GCATTGGTGGTTCAG	5′ end of tRNA-Gly-GCC-3-1
	PMN	18	GCATTGGTGGTTCAGTGG	5′ end of tRNA-Gly-GCC-3-1
		15	GCATTGGTGGTTCAG	5′ end of tRNA-Gly-GCC-3-1
		14	TAGAATTCTCGCCT	Middle of tRNA-Gly-CCC-1-1
	HEK293	15	GCATTGGTGGTTCAG	5′ end of tRNA-Gly-GCC-3-1
		18	GCATTGGTGGTTCAGTGG	5′ end of tRNA-Gly-GCC-3-1
		18	GCATGGGTGGTTCAGTGG	5′ end of tRNA-Gly-GCC-3-1
*M. musculus*	Cerebellum	18	GCATTGGTGGTTCAGTGG	5′ end of tRNA-Gly-GCC-3-1
		15	GCATTGGTGGTTCAG	5′ end of tRNA-Gly-GCC-3-1
		14	CTTCGTGGTCGCCA	Partial 3035a trf-3
	PMN	18	GCATTGGTGGTTCAGTGG	5′ end of tRNA-Gly-GCC-3-1
		17	CATTGGTGGTTCAGTGG	5′ end of tRNA-Gly-GCC-3-1
		15	GCATTGGTGGTTCAG	5′ end of tRNA-Gly-GCC-3-1
	NIH/3T3	18	GCATTGGTGGTTCAGTGG	5′ end of tRNA-Gly-GCC-3-1
		15	GCATTGGTGGTTCAG	5′ end of tRNA-Gly-GCC-3-1
	N2a	18	GCATTGGTGGTTCAGTGG	5′ end of tRNA-Gly-GCC-3-1
		15	GCATTGGTGGTTCAG	5′ end of tRNA-Gly-GCC-3-1
*D. melanogaster*		30	CATCGGTGGTTCAGTGGTAGAATGCTCGCC	5′ end of tRNA-Gly-GCC-3-1
		28	GCATCGGTGGTTCAGTGGTAGAATGCTC	5′ end of tRNA-Gly-GCC-3-1
		17	CCCGGGTTTCGGCACCA	3023 trf-3
*A. thaliana*		15	GGCTAGGTAACATAA	PT-261581 tRF-5
		16	GGGGATGTAGCTCATA	5′ end of tRNA-Ala-CGC-2-1
		16	GGCGGATGTAGCCAAG	PT-218828 tRF-5
*S. cerevisae*		13	GCGGATTTAGCTC	trna9-PheGAA
		13	GCTTCAGTAGCTC	trna19-MetCAT
		28	TCCTTAGTTCGATCCTGAGTGCGAGCTC	tRNA-Cys-GCA-1-1
		29	TCCGTGATAGTTTAATGGTCAGAATGGGC	trna1-AspGTC
*S. pombe*		8	GCTTCAGT	trna49-LeuCAG
		8	GCGGATTT	trna17-SerGCT
		10	CCCTGGGTTC	trna15-AlaTGC

Nucleotide substitutions are in red.

**Table 5 ncrna-08-00034-t005:** Most abundant rRF sequences identified by sRNA-Seq analysis.

		Length (nt)	Sequence
*H. sapiens*	HUVEC	18	TCGTACGACTCTTAGCGG
		19	CTCGTACGACTCTTAGCGG
		18	TCGTACGACTCTTAGCGG
		12	GACTCTTAGCGG
		13	CGACTCTTAGCGG
	PMN	12	GACTCTTAGCGG
		13	CGACTCTTAGCGG
		18	TCGTACGACTCTTAGCGG
	HEK293	12	GACTCTTAGCGG
		13	CGACTCTTAGCGG
		18	TCGTACGACTCTTAGCGG
*M. musculus*	Cerebellum	12	GACTCTTAGCGG
		13	CGACTCTTAGCGG
		25	CAAACGAGAACTTTGAAGGCCGAAG
	PMN	12	GACTCTTAGCGG
		13	CGACTCTTAGCGG
		18	CGATACGACTCTTAGCGG
	NIH	12	GACTCTTAGCGG
		13	CGACTCTTAGCGG
		18	CGATACGACTCTTAGCGG
	N2a	12	GACTCTTAGCGG
		13	CGACTCTTAGCGG
		18	CGATACGACTCTTAGCGG
*D. melanogaster*		11	ACTCTAAGCGG
		12	AACTCTAAGCGG
		30	TGCTTGGACTACATATGGTTGAGGGTTGTA
*A. thaliana*		12	GAGTCTGGTAAT
		14	GGGATGGGTCGGCC
		18	TAGGATAGTGGCCTACCA
*S. cerevisae*		13	TTGACCTCAAATC
		18	TATCTGGTTGATCCTGCC
		19	GCGGCTGTCTGATCAGGCA
*S. pombe*		13	TAAAACTTTCAGC
		13	TTGACCTCAAATC
		24	TTTGACCTCAAATCAGGTAGGACT

## Data Availability

All raw small RNA-seq data generated in this study have been submitted to the NCBI Gene Expression Omnibus under accession number GSE179677. Instructions for Editors and Reviewers: Go to https://www.ncbi.nlm.nih.gov/geo/query/acc.cgi?acc=GSE179677 (accessed on 17 May 2022). Enter token mtyhesicpnunlyv into the box.

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
