# Peer review of "An Expanded Landscape of Unusually Short RNAs in 11 Samples from Six Eukaryotic Organisms"

_ncrna, 2022, doi:10.3390/ncrna8030034_

Round 1
Reviewer 1 Report
The authors present here the identification of unusually short RNAs in samples derived from six organisms. Although in general, the identification (and functional characterization) of new RNA species is of great interest and a (patho)physiological function of at least some of these molecules is likely, the study presented here has several limitations.
Major problems are the lack of replicate samples for the different analyzed sample types as well as the lack of any indication for a physiological relevance of these short RFs identified in this study. The study would provide much more insight if it would focus on a smaller number of organisms and instead would be based on at least replicate samples.
Either a focus on biogenesis of these sRFs (e.g. Dicer depletion) or on function (as e.g. stress response as heat in plants) would be interesting to shed light on the relevance of these short RNAs.
Major concerns:
- Lack of replicates and functional analysis (e.g. comparing non-stressed/stressed, tumor/ non-tumor).
- It is not clear to what extent the identified short RNA fragments are degradation products of the e.g RNA isolation protocol – this could to some point be clarified by comparison of different RNA isolation protocols.
- A significant fraction of the identified short RNAs can presumably not be mapped uniquely to the genome – it is not clear what proportion this is in the different samples. As I understood it correctly, the reads were mapped against type-specific databases. This means that e.g. an identified miRNA-fragment could map as well to other regions of the genome/transcriptome. It would be informative to know, which RNAs were really unambigously mapped.
Minor concerns:
- The first paragraph of the results section is rather hard to read and replicates mainly what is shown in Figure 1. Focusing on some main numbers would increase readibility much.
- The ratio of 12 nt and 13 nt long RNA fragments seem to vary between the different organisms. Does this relates to som physiological relevance or is it reflecting some SNPs that result in altered cleavage pattern? What is the explanation for the higher 13 nt peak in HEK293 compared to a higher 12 nt peak in the other human samples. It is hard to estimate the relevance of these differences without replicate analyses. The same is true for the ratio of 8-15 vs 16-30 nts, it may also be strongly dependent on the type of RNA analyzed (tissue, developmental stage, etc. …) and differences should not be over-interpreted.
- In some instances as on page 4, line 101 the term “unique reads” is used instead of “unique sequence” – this could lead to confusion with “uniquely mapped reads”. In addition to reporting the diversity of certain RNA fractions, also the fraction of reads that could not be uniquely mapped should be reported.
- Is there any explanation that the fraction of 17-19 nt long RNAs is so big in human cells but not in other organisms, including mouse?
- Why it is worth noting that the peak is sharper in mouse and human compared to yeast from an evolutionary perspective?
- Page 4, line 128-134: are these numbers referring to the intersection of the different organisms or of the different RNA types? It is neither clear whether these numbers relate to the interaction or to the type of RNA or organism
- Page 7, lines 174-176: The authors present the most abundant smiRNAs here, however interesting would be to know which of these are conserved (e.g. detected in several samples of an organism or between related organisms as human and mouse)
- page 9, lines 201-203: I agree with a potential use of the semi-miRNAs as biomarkers, however I suggest to rephrase the sentence on the potential function of these, as it is based only on one example.
- Table 1: how were the piRNAs identified as piRNA-fragments when the corresponding parental piRNA is not known?
- For the sdRNAs it seems that single species dominate in the different samples. It would be interesting to see a validation of this e.g. by the presented qPCR method and to see how the expression of the fragments relates to the complete parental snoRNA.
- Table 3: were these uniquely mapped (meaning they could not be mapped to another region of the genome?
- Supplementary Table S3-S13: it is not labeled to what the different tables correspond.
- Figure 4: Why were these miRNAs used for comparison? Not clear why these were chosen and not e.g. the miRNAs with the highest abundance. How was the specificity of the used qPCR method assessed? Some more description on this method should be provided to evaulate the relevance of the presented data
Reviewer 2 Report
The manuscript by Lambert et. al. expands the landscape of short RNAs using small RNA sequencing (sRNA-seq). The authors identified many unusually short RNAs (<16nt RNAs) by reducing the trimming threshold during sRNA library preparations and propose that such measures should be used globally to identify such sRNAs which may be functionally conserved but remained overlooked.
The manuscript “An expanded landscape of unusually short RNAs in 11 samples from six eukaryotic organisms” highlights an important observation of identifying unusually short RNAs of various categories in six eukaryotic organisms. The manuscript is well written, and methodology is very clear.
Major:
Although the RT-qPCR is sufficient to validate the existence of these unusual sRNAs, the only criticism is that there is no evidence of whether these are biologically relevant sRNAs like miR-25 and miR-30a and not just the transcription byproducts or perhaps degradation products of relatively longer sRNAs.
The predominant existence of 12 and 13 nucleotide RNAs rather a smeared existence suggests a regulated production/degradation of these sRNAs and hence an inhibition/knockdown or overexpression experiment studying the biological relevance of any of the identified unusual sRNAs would enhance the outcome of the study. If in the scope of the laboratory, any cell-based assay should suffice.
Minor:
Line 247, 2.3.4 is miswritten as 2.3.3.
Reviewer 3 Report
Dear Authors,
the study you present is of high interest for the scientific community and well designed. The manuscript is well written and sound, the data are presented nicely. I highly recommend publishing the data in the journal. However, just a few minor comments:
(1) Cancer cells yre oftentimes pluripotent. Thus, not every chromosome has exactly 2 copies. If you describe the expression of different RNAs as a model for healthy somatic cells, please consider normalizing their TPMs to the copy number of this gene locus. It is described at ATCC or can be determined by mapping publically availabe DNA or exome data of the cell lines.
(2) You did not allow mismatches at all. What about SNPs that might be encoded in one of the samples? Wouldnt they prevent a certain RNA to be included? Maybe show that you get similar results when allowing mismatches.
(3) Regarding the conservation: Are the selected species in terms of small RNA expression good examples, or accidently not? Maybe do BLAST searches against additional genomes to see if they are highly conserved to close realtives but less conserved to distant species.
Are piRNA clusters (not only single piRNAs) conserved? Are other RNAs also organized in clusters and thus is there a conservation of their synteny?
Round 2
Reviewer 1 Report
- The pooling strategy should be clearly mentioned and referenced in the Material and Methods section. Furthermore, for all samples analyzed, the number of pooled samples should be given, which is at the moment only partially the case.
- In Figure 1A the data points seem not to be correctly assigned to the x-axis. The HEK293 points are alwas a bit right of the respective nt ticks, whereas the HUVEC and PMC points are a little bit shifted to the left. This leaves the impression that the fragments in HEK293 are larger than in the other cells.
- The discussion on the 17-19 nt long RNAs in human (and eventually mouse) samples should be also included in the manuscript.
- Some (especially the rephrased) sentences were hard to read due to the track change mode. Please check the correct phrasing and grammar throughout the manuscript.
Author Response
Review Report (Reviewer 1)
Comments and Suggestions for Authors
- The pooling strategy should be clearly mentioned and referenced in the Material and Methods section. Furthermore, for all samples analyzed, the number of pooled samples should be given, which is at the moment only partially the case.
Response: The pooling strategy, for which we opted, is now clearly mentioned and referenced at the beginning of section 3.4 of the Materials and Methods section, on page 20 of the revised manuscript, as follows: “For primary blood PMN samples involving cell isolation, we used a pooling strategy involving equal mixing of total RNA samples derived from independent biological samples. Pooling of small RNA samples is effective in reducing data variability and reduces the number of replicates, and hence lower cost for the subsequent steps.1 For each of the other samples, a unique total RNA specimen was analyzed.”.
We also now provide the number of pooled samples for all samples analyzed.
(1) Takele Assefa, A., Vandesompele, J., & Thas, O. (2020). On the utility of RNA sample pooling to optimize cost and statistical power in RNA sequencing experiments. BMC Genomics, 21(1), 312. https://doi.org/10.1186/s12864-020-6721-y
- In Figure 1A the data points seem not to be correctly assigned to the x-axis. The HEK293 points are alwas a bit right of the respective nt ticks, whereas the HUVEC and PMC points are a little bit shifted to the left. This leaves the impression that the fragments in HEK293 are larger than in the other cells.
Response: We thank you for raising this issue. The data points were not perfectly aligned so to make each of them visible and distinguishable from other data points. However, we agree that this may be confusing to the readers. We have thus corrected this and modified Figure 1A accordingly in the revised manuscript.
- The discussion on the 17-19 nt long RNAs in human (and eventually mouse) samples should be also included in the manuscript.
Response: We have included the following new paragraph and accompanying references on page 4 of the revised manuscript, as suggested: “The 17 to 19-nt peak (Figure 1) observed in human samples is also present in mouse samples, albeit at a much lower level. These 17 to 19-nt long RNAs mainly comprise sdRNAs, tRFs and rRFs (please refer to Figure 3). In human samples, the 17-nt RNAs seem to be constituted of a unique sequence originating from SNORD30 (please refer to Table 2). It was demonstrated in 2009 that snoRNA-derived RNAs (sdRNAs) from C/D snoRNAs exhibit a bimodal length distribution at ∼17–19 nt and >27 nt and predominantly originate from the 5′ end.4 A possible function of sdRNAs would be to act as a novel source of miRNAs, as a sdRNA originating from the snoRNA ACA45 was found to co-immunoprecipitate with AGO1 and AGO2 in human embryonic kidney 293 cells (HEK293).5 SnoRNAs are also a reported source of piRNAs,6 but their mechanism of action and other of their functions remain unclear. Our data are consistent with 17 to 19-nt RNA species being produced by a processing mechanism preserved in human and mouse, but not in other organisms, which warrants further investigation.”.
(4) Taft, R. J., Glazov, E. A., Lassmann, T., Hayashizaki, Y., Carninci, P., & Mattick, J. S. (2009). Small RNAs derived from snoRNAs. RNA, 15(7), 1233‑1240. https://doi.org/10.1261/rna.1528909
(5) Ender, C., Krek, A., Friedländer, M. R., Beitzinger, M., Weinmann, L., Chen, W., Pfeffer, S., Rajewsky, N., & Meister, G. (2008). A Human snoRNA with MicroRNA-Like Functions. Molecular Cell, 32(4), 519‑528. https://doi.org/10.1016/j.molcel.2008.10.017
(6) He, X., Chen, X., Zhang, X., Duan, X., Pan, T., Hu, Q., Zhang, Y., Zhong, F., Liu, J., Zhang, H., Luo, J., Wu, K., Peng, G., Luo, H., Zhang, L., Li, X., & Zhang, H. (2015). An Lnc RNA (GAS5)/SnoRNA-derived piRNA induces activation of TRAIL gene by site-specifically recruiting MLL/COMPASS-like complexes. Nucleic Acids Research, 43(7), 3712‑3725. https://doi.org/10.1093/nar/gkv214
- Some (especially the rephrased) sentences were hard to read due to the track change mode. Please check the correct phrasing and grammar throughout the manuscript.
Response: We agree that the track changes mode makes it very hard to read the manuscript. We have confirmed the correct phrasing and grammar of the entire manuscript, and will verify the proofs when asked for.
We wish to thank you for your constructive comments and criticisms. We hope that we have addressed your concerns satisfactorily and that you will deem the revised version of our manuscript now acceptable for publication in Non-Coding RNA.